# Novel Phototransformable Fluorescent Protein SAASoti with Unique Photochemical Properties

**DOI:** 10.3390/ijms20143399

**Published:** 2019-07-11

**Authors:** Ilya D. Solovyev, Alexandra V. Gavshina, Alexander P. Savitsky

**Affiliations:** 1Department of Chemistry, M.V. Lomonosov Moscow State University, Leninskie Gory 1/3, Moscow 119991, Russia; 2A.N. Bach Institute of Biochemistry, Research Center of Biotechnology of the Russian Academy of Sciences, Leninsky Ave. 33, bld. 2, Moscow 119071, Russia

**Keywords:** fluorescent proteins, photoconversions of fluorescent proteins, reversible photoswitching

## Abstract

SAASoti is a unique fluorescent protein (FP) that combines properties of green-to-red photoconversion and reversible photoswitching (in its green state), without any amino acid substitutions in the wild type gene. In the present work, we investigated its ability to photoswitch between fluorescent red (‘on’) and dark (‘off’) states. Surprisingly, generated by 400 nm exposure, the red form of SAASoti (R1) does not exhibit any reversible photoswitching behavior under 550 nm illumination, while a combination of prior 470 nm and subsequent 400 nm irradiation led to the appearance of another—R2—form that can be partially photoswitched (550 nm) to the dark state, with a very fast recovery time. The phenomenon might be explained by chemical modification in the chromophore microenvironment during prior 470 nm exposure, and the resulting R2 SAASoti differs chemically from the R1 form. The suggestion is supported by the mass spectrometry analysis of the tryptic peptides before and after 470 nm light exposure, that revealed Met164 oxidation, as proceeds in another dual phototransformable FP, IrisFP.

## 1. Introduction

Phototransformable fluorescent proteins from *Anthozoa* organisms are successfully used in fluorescent microscopy of living systems [1,2,3,4]. Several types of phototransformation can be distinguished: Photoactivation (PA)—the process of irreversible light-induced transformation from a non-fluorescent to a fluorescent form, (paGFP [5], paRFP [6], PAmCherry [7], PAmKate, etc.); green-to-red photoconversion (PC)—irreversible conversion between two fluorescent states with distinct emission colors, as the process involves the extension of the chromophore from a two-ring GFP-like form to a three-ring one, (EosFP [8], Kaede [9], Maple [10], dendFP [11], PSmOrange [12]); ‘on’/‘off’-switching—reversible photoswitching between a fluorescent (‘on’) and a non-fluorescent (‘off’) states, (Dronpa [13], mTFP0.7 [14], rsEGFP [15], Dreiklang [16,17], KFP [18]). While PA and PC are irreversible photochemical processes, the off/on-switching can be repeated several times with the same sample. There are some genetically engineered fluorescent proteins (FPs), which unite PA and PC properties—IrisFP [19], NijiFP [20], and pcDronpa [21]. Certain photophysical and photochemical properties of fluorescent dyes and FPs gave rise to the appearance of different super-resolution localization methods; e.g., they are the basis of Photo-Activated Localization Microscopy (PALM) methods, and observation of stochastic FP blinking—Super-resolution Optical Fluctuation Imaging (SOFI) methods. 

SAASoti was originally discovered in the coral *Stylocoeniella armata* as an irreversibly green-to-red photoconvertible FP [22]. Green-to-red PC occurs under 400 nm illumination and is accompanied by a peptide bond break, as it proceeds in Kaede-like proteins [11]. A 3D tetrameric model of SAASoti was constructed based on the known Kaede structure (PDB ID: 2GW3) [23]. While the wild type SAASoti exists as a tetramer tending towards aggregation, a V127T point mutation at its hydrophobic interface resulted in the monomeric variant of SAASoti [23]. From here on, we will describe this monomeric form—V127T SAASoti. Recently the green form of SAASoti was found to be reversibly switched ‘off’ by irradiating the sample at 470 nm without its PC to the red form [24]. 10 min of 470 nm exposure led to an increase in absorption at 395 nm (chromophore protonated form) and a decrease at 509 nm (chromophore anionic form). The phenomenon is most likely caused by *cis*-*trans* isomerization of the chromophore associated with the change of its protonation state, as it was demonstrated for several reversibly photoswitchable FPs [25].

Interestingly, the F173S point mutation in the case of the photoconvertible FP EosFP led to the appearance of its photoswitching phenomenon (+F191L resulting in IrisFP (Figure 1, blue)) [19], whereas SAASoti is already photoswitchable with phenylalanine occupying the corresponding position—178 (Figure 1). In the case of Dendra2, M159A or F173S mutations caused the same effect. The latter variant had more effective photoswitching to the ‘off’-state and was named NijiFP [20].

Another way to obtain FPs with dual phototransformation nature is to force already photoswitchable proteins to become also photoconvertible, as it was done with Dronpa—a classic photoswitchable FP—that became pcDronpa after C62H/N94S/N102I/E218G/V60A (Figure 1, yellow) mutations [21], where C62H is the key point mutation, as all Kaede-like pcFPs carry conserved the H-Y-G chromophore folding triad. In other words, SAASoti is the first wild type FP described to date possessing both green-to-red PC and reversible photoswitching.

## 2. Results

### 2.1. Photoswitching of the Green Form of SAASoti 

The ‘off’-switching process can be reversed either by illumination at 400 nm or during thermal relaxation. Previously, we investigated thermal relaxation and calculated corresponding constants (τ_1/2_ = 30 min at 25 °C) as SAASoti can be irreversibly converted into the red fluorescent form by 400 nm illumination at pH 7.5 [24]. Figure 2 shows that ‘off’-switching with subsequent 400 nm illumination of the probe can be repeated with the same sample several times without PC to the red form at pH 9.2 (pH-dependency of SAASoti PC was done previously [23]). Green fluorescence photoswitching to the dark state (Figure 2A) at 470 nm (160 mW/cm^2^) can be described by a bi-exponential model (Appendix A). Typically, different types of monomolecular parallel, sequential reactions and their combinations can be described as the sum of the exponential functions (Equation (1)). In the case of the green form photoswitching, the first exponent is the same through all the cycles and is equal to k_1_ = 7.4 ± 0.5 × 10^−3^ s^−1^. The second exponent at the first bleaching cycle has an inverse pre-exponential value that can mean an increase in the fluorescence at the first step with k_2_ = 1.4 ± 0.2 × 10^−2^ s^−1^, and indicates on photochemical transformations. Subsequent ‘off’-switchings have two exponents with the same character—k_1_ = 7.9 ± 0.5 × 10^−3^ s^−1^ and k_2_ = 2.6 ± 0.2 × 10^−2^ s^−1^. ‘On’-switching agrees with a mono-exponential fluorescence growth (Appendix A) with k_growth_ = 0.44 ± 0.02 s^−1^ at 5.7 mW/cm^2^ of 400 nm irradiation power (Figure 2A).
(1)$$I\left(t\right)=C+\overset{n}{\underset{i=1}{\sum }}A_{i}\times \exp\left(-k_{i}t\right)$$

In other words, when going from the first ‘off’-switching cycle to the subsequent ones, the difference in the fluorescence decrease course can be noted. The decrease in the absolute fluorescence intensity between the cycles (Figure 2A) might be caused by the photodestruction of the sample both by 470 nm (during ‘off’-switching) and by illumination at 400 nm (during ‘on’-switching) or because of different extinction coefficients of photochemically transformed SAASoti. Green SAASoti forms before and relaxed after 470 nm exposure are marked G1 and G2, respectively.

### 2.2. Green-to-Red Photoconversion of SAASoti

For different photoconvertible FPs, green-to-red PC was shown to proceed incompletely [20]. As it was demonstrated earlier [23], formation of the red form in the case of SAASoti is accomplished by its subsequent photobleaching under 400 nm. Increasing the 400 nm exposure time in the case of SAASoti leads to the photodestruction of both forms [22]. In other words, some amount of the green form is always present. Interestingly, that combining of prior photobleaching of G1 SAASoti at 470 nm over 10 min (generation of G2 form) with subsequent illumination at 400 nm resulted in accelerated photoconversion (Figure 3A).

The idea was to shift the equilibrium between the neutral and anionic states of the chromophore and make it pre-adopted for the PC, as only the neutral chromophore in Kaede-like proteins can be photoconverted to the red form under exposure to ultraviolet light [11]. Dendra2 is an interesting FP that can be photoconverted either by 400 nm illumination or 488 nm light (with much less efficiency) [11], making primed conversion possible in this case [26]. As the G2 form instantly returns from the ‘off’ state during 400 nm illumination, while green-to-red photoconversion occurs within several minutes, we suggest that some chemical modification of the amino acid residues in the vicinity of the chromophore might take place. In other words, these two forms of SAASoti—G1 and pre-irradiated at 470 nm, G2—are supposed to differ chemically. SAASoti converted by the ‘original’ technique during exposing the sample to 400 nm light we will mark R1, and with prior 470 nm irradiation of the green form—R2. The process is described by a three-exponential function, where k_1_ and k_2_ are responsible for the red form generation and k_3_—for its irreversible photodestruction (Table 1). There is almost no difference in the initial rate value in the red SAASoti formation (Figure 3B) when an old SAASoti sample (storage at 4 °C for about 6 months without any reductive agent) and pre-irradiated samples were studied. The data is in a good agreement with the chemical modification suggestion. 

### 2.3. Photoswitching of the Red SAASoti Form

We also investigated if the red form of SAASoti—R1 and R2—could be photoswitched from their fluorescent ‘on’ to the ‘off’-states and whether this process was reversible. Firstly, we recorded kinetics of the red fluorescence decrease of R1 under 550 nm illumination (300 mW/cm^2^) during 60 min (by contrast, G1 can be completely ‘off’-switched within 10 min [24]). The process is irreversible as no thermal relaxation or ‘off’-to-‘on’ light-induced photoswitching were observed. The absorbance spectra before and after 550 nm exposure also did not reveal the presence of the chromophore protonated form (as it was observed in the case of rIrisFP, [19]). The data is not presented. Fluorescence decay during 550 nm exposure in the case of R1 is most likely caused by the protein’s photodestruction (Appendix A, black line). 

On the contrary, when irradiating R2 SAASoti with 550 nm light we observed some ‘on’-‘off’ photoswitching of this form. It is important to note, that 550 nm exposure time should not exceed 50–60 s for R2 SAASoti as a second (photodestruction) component dominates in this case. ‘Off’-switched R2 form can relax within 10 min to its fluorescent ‘on’-state (G1 relaxes during more than 1 h). Moreover, this ‘on’-‘off’ switching can be repeated several times (Figure 4A). It should be noted, that the fluorescence decrease is no more than 20% of the initial intensity value, while in the case of G1 fluorescence can be completely switched ‘off’. Interestingly, when investigating an ‘old’ SAASoti sample (storage at 4 °C for about 6 months without any reductive agent) under 550 nm irradiation for 50 s and subsequent 10 min thermal relaxation we also observed some photoswitching (Figure 4B); no more than 10% of initial intensity for R1 in the case of the ‘old’ sample.

Absorbance spectra of the R2 form before and after 550 nm illumination during 50 s revealed the decrease of 573 nm maxima and a minor peak broadening in the range of 400–500 nm (Figure 5A). Spectral changes could be presented more clearly by the subtraction of the initial R2 absorbance spectrum from the spectrum of the exposed to 550 nm light protein (Figure 5B). Increasing absorbance at 400–500 nm is probably caused by the protonation of the red form, but because of the small extinction coefficient of the protonated form and superposition with the green anionic spectra, the effect is barely visible.

The red fluorescence decrease is described by a bi-exponential function (Equation (1)), where the first component is responsible for a reversible photoswitch and the second one—for the photodestruction (Appendix A, Table 2).

Taking into account all data on SAASoti phototransformations and experiments on ‘old’ samples received so far, we can suggest that there exists a difference between chromophore environments in R1 and R2 SAASoti forms. Preliminary exposure to 470 nm might cause chemical modification that promotes reversible photoswitching (to some extent) of the red form of SAASoti. The phenomenon of the amino acid residue’s chemical modification during blue light exposure was shown previously when studying another dual phototransformable protein IrisFP [27]. The authors demonstrated Met159 oxidation. In the case of SAASoti, methionine occupies the same position: 164 (Figure 1). MALDI-TOF/TOF was used to identify the difference between G1 and G2 forms that might occur after 470 nm exposure and might lead to the reversible photoswitching of the R2 form. As can be seen from Appendix A, 10 min exposure to 470 nm light leads to the appearance of a 1466.7 Da peptide from a 1450.7 Da one, but the latter is still present in the irradiated sample, indicating an incomplete photo-oxidation reaction (see Appendix A). The 16 Da mass shift is caused by the methionine 164 oxidation, as it follows from the fragmentation spectrum of 1466.7 Da peptide (Appendix A). Thus, we succeeded to identify one of the possible photochemical modifications in the close vicinity of the chromophore. On this basis, we suggest the main scheme (Scheme 1) of possible transformations between different forms of SAASoti.

## 3. Discussion

Photoconvertible and photoswitchable (in their green form) FPs pcDronpa and pcDronpa2 do not possess any red form switching [21]. By comparing this fact with the data received on the red form of IrisFP [19], that can be completely ‘off’-switched, the authors [21] suppose that the phenomenon can be explained by the reduced rotational freedom of the chromophore in the case of pcDronpa2. The switching efficiency in the case of NijiFP is equal to ~95% (green form) and ~80% (red form) [20].

Unlike G1, R1 SAASoti does not exhibit any significant reversible photoswitching. Only generated from preliminarily ‘off’ switched green form—R2 shows limited photoswitching under 550 nm exposure for 50 s, and increasing the exposure time leads to the irreversible photodestruction. There are at least two suggestions for explaining the phenomenon. R2 SAASoti in its ‘off’-state might have a lower pKa value than that of the green dark state. The pKa shift may appear by position changing of the nearest amino acid residues in comparison with the green form, or photochemical modification of an amino acid during photoconversion or photoswitching. Dark states are stabilized by the tetrad Arg66-Ser142-His194-Glu212 (Figure 1, green) in most switchable and multi-photochromic proteins [19,20].

In the case of IrisFP, the red form crystal structure is resolved as a *cis-* and *trans-*form mixture [19]. We suppose that the red state of SAASoti undergoes partial twisting with a fast recovery time. In spite of the fact that all GFP-like proteins share common structure and key amino acid residues, the unique nature of the chromophore microenvironment has a direct impact on the FP’s properties, being the cause of such a distinct diversity in the GFP-family and even in the subfamilies (e.g., Kaede-like proteins). SAASoti is an interesting example of when a combination of properties do not exactly fit in the proposed conceptions of FP phototransformations, and further investigations on this protein may shed light on its unique phototransformative nature.

## 4. Materials and Methods

### 4.1. Protein Expression and Purification

V127T SAASoti was overexpressed in *E. coli* BL21 (DE3) as described previously [23]. Protein purity was analyzed by gel-electrophoresis. 

### 4.2. Absorbance and Fluorescence Measurements

Absorption spectra were measured in the range 250–700 nm using a Shimadzu UV-1650 spectrophotometer (Kyoto, Japan) and a 3 mm quartz micro-cuvette (Hellma, Mülheim, Germany). Samples were dissolved in 20 mM Tris-HCl, 150 mM NaCl, pH 7.5 buffer if not stated otherwise. A Cary Eclipse spectrofluorimeter (Varian, Mulgrave, Australia) with a 3 mm quartz microcuvette was used for measurements for excitation in the range 250–600 nm and for fluorescence in the range 450–700 nm. Excitation and emission slits were set to 5 nm. Fluorescence spectra, photoconversion and photobleaching kinetics of the red form of the protein were obtained on a fiber optic SpectrClaster (Moscow, Russia) spectrometer in the spectral range 400–900 nm, as described previously [24] and on a Cary Eclipse fluorescence spectrometer (Varian, Mulgrave, Australia) in the spectral range 450–750 nm. The kinetic experiments were repeated as least three times. Kinetic fits and spectra were performed using Origin 8.5 software.

### 4.3. Mass Spectrometry

The sample of G2 SAASoti for mass-spectrometry was obtained in cuvette at 470 nm (160 mW/cm^2^) over 10 min. The subsequent sample preparation of G1 and G2 solutions was performed as it was describer earlier [23]. Mass spectra were obtained using a MALDI-TOF Ultraflextreme BRUKER mass spectrometer (Bruker Daltonics, Bremen, Germany). Mass spectra were analyzed using FlexAnalysis 3.3 software (Bruker Daltonics, Bremen, Germany). Comparison of experimentally determined and calculated masses of tryptic peptides of SAASoti were performed using Mascot software (www.matrixscience.com). Searches were performed by taking into account the possible oxidation of methionine and cysteine residues, or glutamate decarboxilation by 470 nm irradiation and formation of a heterocycle in the fluorophore. In order to verify the presence or absence of modifications we obtained fragmentation spectra for distinct peptides (1466.7 Da). Combined analysis of the MS + MS/MS results was performed with Biotools 3.2 (Bruker Daltonics) software. Combined analysis of the MS + MS/MS results was performed with Biotools 3.2 (Bruker Daltonics) software.

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
