# Peer review of "Novel Phototransformable Fluorescent Protein SAASoti with Unique Photochemical Properties"

_ijms, 2019, doi:10.3390/ijms20143399_

Round 1

Reviewer 1 Report

The Manuscript by Solovyev et al. reports on the unexpected photochemical reactions of the monomerized V127T version of recently discovered SAASoti fluorescent protein. Novel photoactive fluorescent proteins should be put under meticulous scrutiny as potential sources of novel chromophore modifications, which often lead to new microscopy techniques. The topic itself is of great interest to the field, and the data in this Manuscript merit publication. The wild-type SAASoti exhibits a range of photochemical reactions, resembling the ones of engineered IrisFP.

The Referee would like to share the following comment on the Manuscript:

Proofreading of the Manuscript is necessary.  Besides some incomplete and grammatically incorrect sentences, there are seven places with ‘Error!’ text instead of the reference, confusing labels in tables, etc.

The authors mention that ‘The absorbance spectra before and after 550 nm exposure also did not reveal the presence of the chromophore protonated’. What spectra are the Authors referring to? Fig1 show only the ones for 400-470nm illumination cycle. 

The same question for the sentence  ‘as residual absorbance still can be seen from the switched ‘off’ spectra.’ 

Table 2. Where are the units?

Figure 4. The use of ‘power, %’ is inappropriate, please provide the plots in the ‘W per cm2’ coordinate.

Figure 4. Please include the regression in order to assess the deviation from linearity

Table 4. Truly enigmatic one. The Referee has spent quite some time thinking about ‘At ? nm’, positional coding of the hyphens in consequent rows, A1/A2 (4→1), A1/A2 (-3), semantic pairs ‘On’-switching : Switching on, ‘Off’-switching : Switching off. Please amend the table

In the opinion of the Referee, the terms  ‘photoconversion’,  ‘photoswitching,’ and ‘photoactivation’  are overused in the FP-related literature and do not help the general reader the tiniest bit in understanding these complex phenomena. These are all just various photochemical reactions and should be named as such, even in case of cis-trans isomerization of the chromophore (as it involves the rearrangement of the hydrogen bond network). Please consider using terms ‘photochemical reaction’ and ‘photoproduct’ instead in the text, as it may greatly help the general reader. 

The terminology is confusing and inconsistent. One can find  ‘piGreen’ and ‘piRed’  existing in Table 4 only. While the Referee succeeded in understanding the central claims of the paper, the general reader will most likely fail. For the sake of clarity, the Referee urges the authors to simplify the text somehow, for example, by naming the forms as G1,2,3... R1,2,3… and providing the hypothesized chemical scheme 1 at the beginning of the Manuscript. The Referee assures the Authors, that the ‘yellow star’ in the scheme 1 does not help at all in understanding, and following the meaning of  ‘these two forms of SAASoti – gSAASoti and pre-irradiated at 470 nm’ is difficult. Since the chemical identity of the forms is not established, naming them in such an exquisite manner seems inappropriate.

Author Response

Thank You for the comments. We have revised the Manuscript accordingly and uploaded the new version.

1)      Proofreading of the Manuscript is necessary.  Besides some incomplete and grammatically incorrect sentences, there are seven places with ‘Error!’ text instead of the reference, confusing labels in tables, etc.

References were corrected.

2)      The authors mention that ‘The absorbance spectra before and after 550 nm exposure also did not reveal the presence of the chromophore protonated’. What spectra are the Authors referring to? Fig1 show only the ones for 400-470nm illumination cycle. 

The corresponding spectra were added to the manuscript (Fig. 4). There is a minor difference in absorbance spectra between R2 form before and after illumination at 550 nm (50 s), as R2 form is partially photoswitchable to the non-fluorescent form. We suppose that absorbance maximum of R2 protonated form is superposed with green anionic form, as the photoconversion proceeds incompletely. The information is added to the Manuscript.

3)      The same question for the sentence ‘as residual absorbance still can be seen from the switched ‘off’ spectra.’ 

The statement was corrected.

4)      Table 2. Where are the units?

The units were added o the Table 2.

5)      Figure 4. The use of ‘power, %’ is inappropriate, please provide the plots in the ‘W per cm2’ coordinate.

The data were excluded from the Manuscript.

6)      Figure 4. Please include the regression in order to assess the deviation from linearity

The data was excluded from the Manuscript.

7)      Table 4. Truly enigmatic one. The Referee has spent quite some time thinking about ‘At ? nm’, positional coding of the hyphens in consequent rows, A1/A2 (4→1), A1/A2 (-3), semantic pairs ‘On’-switching : Switching on, ‘Off’-switching : Switching off. Please amend the table

The Table 4 was excluded from the Manuscript.

8)      In the opinion of the Referee, the terms  ‘photoconversion’,  ‘photoswitching,’ and ‘photoactivation’  are overused in the FP-related literature and do not help the general reader the tiniest bit in understanding these complex phenomena. These are all just various photochemical reactions and should be named as such, even in case of cis-trans isomerization of the chromophore (as it involves the rearrangement of the hydrogen bond network). Please consider using terms ‘photochemical reaction’ and ‘photoproduct’ instead in the text, as it may greatly help the general reader. 

The terms ‘photoactivation’, ‘photoconversion’, and ‘photoswitching’ are common terminology in the FP-related literature. The corresponding definitions were added to the beginning of the Manuscript. All of the described processes do provide a variety of possible photochemical reactions in the FPs’ chemistry, but we tried to avoid the usage of more generalized terms in order to avoid possible confusions and to describe distinct processes, as SAASoti turned out to have complex phototransformation nature.

9)      The terminology is confusing and inconsistent. One can find  ‘piGreen’ and ‘piRed’  existing in Table 4 only. While the Referee succeeded in understanding the central claims of the paper, the general reader will most likely fail. For the sake of clarity, the Referee urges the authors to simplify the text somehow, for example, by naming the forms as G1,2,3... R1,2,3… and providing the hypothesized chemical scheme 1 at the beginning of the Manuscript. The Referee assures the Authors, that the ‘yellow star’ in the scheme 1 does not help at all in understanding, and following the meaning of  ‘these two forms of SAASoti – gSAASoti and pre-irradiated at 470 nm’ is difficult. Since the chemical identity of the forms is not established, naming them in such an exquisite manner seems inappropriate.

All confused SAASoti forms were replaced according to the proposed designations: gSAASoti – G1, exposed to 470 nm gSAASoti – G2, ‘originally’ photoconverted by 400 nm illumination – R1, with prior 470 nm photoswitching of the green form – R2. The updated Scheme 1 was added to the Manuscript.

We hope the revised version is now suitable for publication and look forward to hearing from you in due course.

Reviewer 2 Report

Sebastian Ilya D. Solovyev et. al. in the submitted manuscript “Novel phototransformable fluorescent protein SAASoti with unique photochemical properties” describe analysis of fluorescent protein SAASoti that combines properties of green-to-red photoconversion and reversible photoswitching (in its green state). Authors have investigated its ability to photoswitch between fluorescent red (‘on’) and dark (‘off’) states: red form, generated by 400 nm exposure, does not exhibit any reversible photoswitching behavior under 550 nm, while combination of prior 470 nm and subsequent 400 nm irradiation leds to the appearance of another form which can be partially photoswitched (550 nm) to the dark state with very fast recovery time. The authors explain these observations by chemical modification in the chromophore microenvironment during prior 470 nm exposure.

Overall, the work is of a good quality and interesting. Thus I am happy to recommend this manuscript to be accepted for publishing in International Journal of Molecular Sciences (MDPI) after major revision, because several additional experiments need to be performed:

Minor points:

1.      Please correct the links to citations, remove “[Error! Bookmark not defined.]” from text and tables.

2.      Please mark all figures panels correctly, sections A and B are not corresponding the legends.

3.      Please provide the correct description of the statistics shown in tables and images. Indicate number of replicates, mention how data are presented (mean ± s.d. or other), fittings should have corresponding residual graphs which indicate the quality of the fitting and etc.

4.      Please explain all abbreviations used in the text: SAASoti, SOFI, PALM, and etc. It makes text more understandable.

Major points:

1.      Authors mention several times the influence of protonation on switching properties of SAASoti. However, they do not provide any experimental data. I suggest to perform switching experiments in the several solution of different pH.

2.      Authors provide several experiments with “fresh” and “old” proteins stored without reducing agent. However, the important control would be to show the same experimental curves with “old” protein stored in presence of reducing agent.

3.      Authors propose chemical modification as possible explanation of switching behavior. However, no direct evidence is provided. It should be supported by at least one experimental data: MS, NMR, X-ray or similar.

Author Response

Thank You for the comments. We have revised the Manuscript accordingly and uploaded the new version.

Minor points

1)      Please correct the links to citations, remove “[Error! Bookmark not defined.]” from text and tables.

The links were corrected.

2)      Please mark all figures panels correctly, sections A and B are not corresponding the legends.

All the figures and corresponding legends were marked correctly.

3)      Please provide the correct description of the statistics shown in tables and images. Indicate number of replicates, mention how data are presented (mean ± s.d. or other), fittings should have corresponding residual graphs which indicate the quality of the fitting and etc.

S.E. values and residual graphs were added to the tables and figures, respectively.

4)      Please explain all abbreviations used in the text: SAASoti, SOFI, PALM, and etc. It makes text more understandable.

SAASoti is the name of the protein, and it was first used in [1], and then it was used in [2] and [3]. Abbreviations for SOFI and PALM were added to the text.

Major points

1)      Authors mention several times the influence of protonation on switching properties of SAASoti. However, they do not provide any experimental data. I suggest to perform switching experiments in the several solution of different pH.

‘On’-‘off’ switching is caused by a cis-trans isomerization of the chromophore excited state and is accomplished by a change of its protonation state.  During isomerization the amino acid network in the vicinity of the chromophore changes and, as a result, the proton transmission chain does, too. [4] Experiments on photoswitching in solutions with different pH will only shift the equilibrium between naturally protonated and anionic forms in the ground state and totally ignore processes in the excited state.

2)      Authors provide several experiments with “fresh” and “old” proteins stored without reducing agent. However, the important control would be to show the same experimental curves with “old” protein stored in presence of reducing agent.

The experiments on the ‘old’ sample were held because of the chemical modification suggestion, and its 6 months storage with DTT or BME seems to be impossible now. Nevertheless, we can suggest that addition of the reducing agent to the already sulfoxidized sample will not reduce Met from Met-O.

3)      Authors propose chemical modification as possible explanation of switching behavior. However, no direct evidence is provided. It should be supported by at least one experimental data: MS, NMR, X-ray or similar

The statement is now supported by the mass spectrometry analysis of the tryptic peptides before and after 470 nm light exposure, that revealed Met164 oxidation (See Fig. S4 and Table S2), as it proceeds in an another dual phototransformable FP IrisFP [5]. We succeeded to identify one of the possible photochemical modification – Met164 – in the close vicinity of the chromophore. The information is added to the Manuscript and SI.

References

1. Lapshin, G.; Salih, A.; et. al. Fluorescence color diversity of great barrier reef corals. J. Innov. Opt. Heal. Sci. 2015, 8 (4), 1550028.

2. Solovyev, I.D.; Gavshina, A.V.; et al. Monomerization of the photoconvertible fluorescent protein SAASoti by rational mutagenesis of single amino acids. Sci. Rep. 2018, 8, 15542.

3. Solovyev, I.; Gavshina, A.; Savitsky, A. Reversible photobleaching of photoconvertible SAASoti-FP. J. Biomed. Photon. Eng. 2017, 3(4), 040303-1. doi:10.18287/JBPE17.03.040303.

4. Coquelle, N.; Sliwa, M. et. al. Chromophore twisting in the excited state of a photoswitchable fluorescent protein captured by time-resolved serial femtosecond crystallography. Nature Chemistry. 2018, 10, pages 31–37

5. Duan, C.; Adam, V.; et. al. Structural evidence for a two-regime photobleaching mechanism in a reversibly switchable fluorescent protein. J. Am. Chem. Soc. 2013, 135(42):15841-50. doi: 10.1021/ja406860e.

We hope the revised version is now suitable for publication and look forward to hearing from you in due course.

Round 2

Reviewer 2 Report

Sebastian Ilya D. Solovyev et. al. in the re-submitted manuscript “Novel phototransformable fluorescent protein SAASoti with unique photochemical properties” have addressed some of my concerns. I would prefer if these all of them are addressed properly before publishing the final version in International Journal of Molecular Sciences (MDPI):

1) Authors mention several times the influence of protonation on switching properties of SAASoti. However, they do not provide any experimental data. I suggest to perform switching experiments in the several solution of different pH.

Authors reply: ‘On’-‘off’ switching is caused by a cis-trans isomerization of the chromophore excited state and is accomplished by a change of its protonation state.  During isomerization the amino acid network in the vicinity of the chromophore changes and, as a result, the proton transmission chain does, too. [4] Experiments on photoswitching in solutions with different pH will only shift the equilibrium between naturally protonated and anionic forms in the ground state and totally ignore processes in the excited state.

Comment on reply: Please add this explanations and reference to the main text of the manuscript as it is important information which reader needs to know!

2) Authors provide several experiments with “fresh” and “old” proteins stored without reducing agent. However, the important control would be to show the same experimental curves with “old” protein stored in presence of reducing agent.

Authors reply: The experiments on the ‘old’ sample were held because of the chemical modification suggestion, and its 6 months storage with DTT or BME seems to be impossible now. Nevertheless, we can suggest that addition of the reducing agent to the already sulfoxidized sample will not reduce Met from Met-O.

Comment on reply: I do not understand why it is not possible to perform this experiment? Are authors in a rushing the publication? Please explain. In my opinion, it the suggested experiment would be a nice piece of information which will support the conclusions.

3) Authors propose chemical modification as possible explanation of switching behavior. However, no direct evidence is provided. It should be supported by at least one experimental data: MS, NMR, X-ray or similar

Authors reply: The statement is now supported by the mass spectrometry analysis of the tryptic peptides before and after 470 nm light exposure, that revealed Met164 oxidation (See Fig. S4 and Table S2), as it proceeds in an another dual phototransformable FP IrisFP [5]. We succeeded to identify one of the possible photochemical modification – Met164 – in the close vicinity of the chromophore. The information is added to the Manuscript and SI.

Comment on reply: The provided mass spectrum in Fig. S4 shows very low abundance of M + 16 ion. In fact it looks like a peak of noise. Please provide evidence that peak is significant compared to other peaks which could be found in the baseline.